# Squashing and supersymmetry enhancement in three dimensions

**Joseph Minahan, Usman Naseer and Charles Thull**

*Department of Physics and Astronomy, Uppsala University,*
*Box 516, SE-751 20 Uppsala, Sweden*

{`joseph.minahan,usman.naseer,charles.thull`}`@physics.uu.se`

## Abstract

We consider mass-deformed theories with $\mathcal{N} \geq 2$ supersymmetry on round and squashed three-spheres. By embedding the supersymmetric backgrounds in extended supergravity we show that at special values of mass deformations the supersymmetry is enhanced on the squashed spheres. When the $3d$ partition function can be obtained by a limit of a $4d$ index we also show that for these special mass deformations only the states annihilated by extra supercharges contribute to the index. By using an equivalence between partition functions on squashed spheres and ellipsoids, we explain the recently observed squashing independence of the partition function of mass-deformed ABJ(M) theory on the ellipsoid. We provide further examples of such simplification for various $3d$ supersymmetric theories.

## 1. Introduction and summary

Localization is a powerful tool with a broad range of applications in supersymmetric field theories [1]. In general one can put supersymmetric theories on the round sphere $\mathbb{S}^d$ while preserving the same amount of supersymmetry as the theory on flat Euclidean space $\mathbb{R}^d$. Localization then gives the free energy of the theory in terms of finite dimensional matrix model integrals which depend on various parameters of the theory such as couplings $\vec{\tau}$ and masses $\vec{m}$. With enough supersymmetry the matrix models can be very simple, e.g., the Gaussian matrix model one finds for $\mathcal{N} = 4$ super Yang-Mills [2].

It is still possible to localize the supersymmetric theory after squashing the sphere, that is deforming the metric by a parameter $b$, where $b=1$ corresponds to the round sphere [3–6]. The metric deformation breaks the supersymmetry, but it can be partially restored by turning on appropriate background fields in the supergravity multiplet à la [7]. The resulting matrix models after localization are in general more complicated than for the round sphere, but contain much more information. For example the derivatives of the free energy $\mathcal{F}(b, \vec{\tau}, \vec{m})$ with respect to the deformation parameter, couplings and masses give new constraints between various integrated or partially integrated correlators of the supersymmetric theory. Using holographic duality, these relations imply non-trivial constraints on string theory amplitudes and can be used to fix protected terms in string theory effective actions [8–13].

A convenient situation arises when the free energy is independent of the squashing parameter. In this case one can obtain an infinite number of relations between correlators without having to deal with the complicated matrix models. This can happen in two distinct ways: (1) The free energy is independent of the squashing without tuning any other parameters of the theory. The free energies of this type are the $3d$ and $4d$ theories with $\mathcal{N} = 2$ supersymmetry placed on squashed spheres with $SU(2) \times U(1)$ isometry, as studied in [3, 14]. (2) The free energy becomes independent of the squashing when certain parameters are tuned to a special value. For example, it was pointed out in [15, 16] that the free energy of $4d$ $\mathcal{N} = 2$ theory on the squashed sphere with an adjoint hypermultiplet of a special mass is independent of the squashing. This leads to an infinite number of relations between (integrated) correlators of $\mathcal{N} = 4$ SYM [16], some of which were obtained earlier in [13].

A similar phenomenon was discovered by [17] for ABJ(M) theory. The free energy on the squashed sphere becomes squashing-independent if one deforms by a special mass corresponding to a Cartan of the flavor symmetry $SO(2) \subset SO(4) \times U(1)$. In this case the simplification appears miraculous as there are complicated cancelations between different contributions to the partition function which only become apparent after a non-trivial rewriting of the partition function using the Cauchy determinant formula. We are thus led to search for the underlying reason for this simplification. In this paper we study and explain this simplification for supersymmetric theories on squashed three-spheres. A companion paper [18] analyzes $\mathcal{N} = 2^*$ theory on general four-manifolds. The general strategy is to show that there is enhanced supersymmetry at this special value of the mass.

Along the way we find various $3d$ theories whose free energy is squashing-independent. The partition function can be assembled in terms of a classical piece and one-loop determinants associated with the vector- and matter-multiplets. From these ingredients we show that free energy of various theories of an $\mathcal{N} = 4$ vector-multiplet with an adjoint hypermultiplet of mass[1] $\mu = +\mathrm{i}\frac{b+b^{-1}}{2}$ is independent of $b$. This follows directly from the cancellation between one-loop determinants. Similarly, we show that the free energy of ABJ(M) theory deformed by a mass $\mu = +\mathrm{i}\frac{b+b^{-1}}{2}$ corresponding to a Cartan of the flavor symmetry is squashing-independent. This, again, follows after a non-trivial rewriting of the partition function that uses the Cauchy determinant formula[2]. A further example is we discuss is the so called $N_f$-model, showing squashing independence for $\mu = \mathrm{i}\frac{b+b^{-1}}{2}$ as well as $\mu = \pm\mathrm{i}\frac{b-b^{-1}}{2}$.

To understand the squashing independence, our main tools of analysis are the results of [19, 20] and the relations between $4d$ indices and $3d$ partition functions. For the first kind of squashing independence, it is sufficient to consider the transversely holomorphic foliation (THF) that deter-

---

[1]One can add supersymmetric mass terms by coupling the matter-multiplet to a background $U(1)$ vector-multiplet. Here, by mass we mean the constant value of the scalar field in this background multiplet.

[2]The simplification in [17] occurs for $\mu = \pm\mathrm{i}\frac{b-b^{-1}}{2}$.

mines the partition functions of the supersymmetric theories. We show that this structure is the same for the round sphere and the $SU(2) \times U(1)$-deformed sphere of [3]. The $4d$ index which gives the relevant partition function depends on a single fugacity. For the second kind of simplification, we show that there is enhanced supersymmetry at the special mass values. For $\mu = +\mathrm{i}\frac{b+b^{-1}}{2}$ there are four extra supercharges, while for $\mu = \pm\mathrm{i}\frac{b-b^{-1}}{2}$ there are two extra supercharges. The enhanced supersymmetry is found by embedding the $3d$ $\mathcal{N} = 2$ supergravity in appropriate $3d$ extended supergravity. For $\mu = \pm\mathrm{i}\frac{b-b^{-1}}{2}$, the THF associated with the extra supercharge coincides with the THF for the round sphere. From this one deduces that the partition function is independent of the squashing. Combining this with the equivalence of partition functions between ellipsoids and squashed spheres, one is led to the squashing independence in [13]. On the other hand, the supersymmetry enhancement for the $\mu = \mathrm{i}\frac{b+b^{-1}}{2}$ point does not explain the squashing independence. However based on the triviality of the $\mathcal{N} = 4$ partition function at this point we suspect that the theory becomes topological.

The rest of this paper is structured as follows: In section 2 we review general features about supersymmetric partition functions on three manifolds and analyze the THF on the round sphere, the $U(1) \times U(1)$ symmetric ellipsoid and the $SU(2) \times U(1)$ symmetric squashed sphere. In section 3 we show that the supersymmetry is enhanced at the special values of the mass parameters by considering a twisted dimensional reduction of $4d$ $\mathcal{N} = 2$ supergravity which we discuss in appendix B. The corresponding analysis for $3d$ $\mathcal{N} = 6$ conformal supergravity is presented in appendix C which explains the symmetry enhancement for ABJ(M). In section 4 we explain the squashing independence and the symmetry enhancement by studying the $4d$ indices which give $3d$ partition functions upon compactification. Appendix A contains details of the simplification of partition functions for SYM, including the $N_f$ model, and ABJ(M) at special mass parameters.

## 2. $\mathcal{N} = 2$ supersymmetry and partition functions on deformed spheres

In the literature two distinct deformations of the three-sphere have been considered. The first deformation, which we refer to as the ellipsoid, preserves only a $U(1) \times U(1)$ isometry. Supersymmetry on the ellipsoid as well as localization of path integrals were discussed in [3, 21]. The second deformation, which we refer to as the squashed sphere, preserves an $SU(2) \times U(1)$ isometry. There are two distinct possibilities for how to preserve some supercharges on this metric background [3, 4, 22] and they lead to distinct results for the partition function which is equal to the round sphere partition function in one case and the ellipsoid partition function in the other case. The purpose of this section is to explain this using the techniques and results from [19, 20].

On a Riemannian three manifold $\mathcal{M}$, the metric will break all supersymmetries. To preserve some (rigid) supersymmetry one then has to turn on additional background fields in the supergravity multiplet [7]. For new minimal supergravity these fields are a gauge field $A$ for the $U(1)_R$ symmetry, a dual field strength $V$ for the graviphoton and a scalar field $H$. Then every spinor $\zeta, \widetilde{\zeta}$ solving the

Killing spinor equations[3]

$$\nabla_\mu \zeta - \mathrm{i}\left(A_\mu + V_\mu\right)\zeta - H\gamma_\mu \zeta + \frac{1}{2}\varepsilon_{\mu\nu\rho}V^\nu\gamma^\rho\zeta = 0,$$

$$\nabla_\mu \widetilde{\zeta} + \mathrm{i}\left(A_\mu + V_\mu\right)\widetilde{\zeta} - H\gamma_\mu \widetilde{\zeta} - \frac{1}{2}\varepsilon_{\mu\nu\rho}V^\nu\gamma^\rho\widetilde{\zeta} = 0\,,$$

(2.1)

corresponds to a supercharge preserved by the background.

Equivalent to the existence of at least one solution to the equations in (2.1) is that the manifold $\mathcal{M}$ admits a transversely holomorphic foliation (THF) compatible with the metric. Assuming that $\mathcal{M}$ is oriented, such a THF is given by a nowhere vanishing, normalized vector field $\xi$ subject to the constraint that the tensor $\Phi^\mu{}_\nu \equiv \varepsilon^\mu{}_{\nu\rho}\xi^\rho$ satisfies the integrability condition

$$\Phi^\mu{}_\rho \mathcal{L}_\xi \Phi^\rho{}_\nu = 0\,. \tag{2.2}$$

The orbits of $\xi$ define leaves of a one-dimensional oriented foliation of $\mathcal{M}$ and $\Phi$ induces an almost complex structure on the normal bundle of the foliation. Due to the integrability condition the almost complex structure is constant along the orbits of $\xi$. As a consequence, one can construct local coordinates $(\tau, z, \bar{z})$ such that $\xi$ is the vector

$$\xi = \partial_\tau\,, \tag{2.3}$$

and the metric can be written as

$$ds^2 = \left(\mathrm{d}\tau + h\left(\tau, z, \overline{z}\right)\mathrm{d}z + \overline{h}\left(\tau, z, \overline{z}\right)\mathrm{d}\overline{z}\right)^2 + c\left(\tau, z, \overline{z}\right)^2 \mathrm{d}z\mathrm{d}\overline{z}. \tag{2.4}$$

For later reference we note that in such adapted coordinates the tensor $\Phi$ takes the nice form

$$\Phi^\mu{}_\nu = \begin{pmatrix} 0 & \mathrm{i}h & -\mathrm{i}\overline{h} \\ 0 & -\mathrm{i} & 0 \\ 0 & 0 & \mathrm{i} \end{pmatrix}. \tag{2.5}$$

The metric on the normal bundle of the foliation is hermitian, hence (2.4) is referred to as the transversely hermitian metric (THM) compatible with the foliation. Because of ambiguities, a THF with a compatible THM does not correspond to a unique set of background fields. However, it is sufficient to write down supersymmetric Lagrangians which, among other parameters, depend on the THF, the THM and the aforementioned ambiguities.

Three-sphere and its deformations that we consider preserve at least two supercharges of opposite R-charge, hence there exists a pair $\zeta, \widetilde{\zeta}$ of Killing spinors. From these Killing spinors we can

---

[3]The gamma matrices in 3 dimensions are the Pauli matrices.

construct the Killing vector[4]

$$K = \zeta \gamma^\mu \widetilde{\zeta} \partial_\mu. \tag{2.6}$$

Assuming that $K$ generates a single isometry, we can write the normalized vector field as $\xi = \Omega^{-1} K$, where $\Omega$ is the norm of $K$. We can now define coordinates $(\psi, z, \overline{z})$ where $K = \partial_\psi$ and the metric takes the form

$$\mathrm{d}s^2 = \Omega(z, \overline{z})^2 \left(\mathrm{d}\psi + a(z, \overline{z})\,\mathrm{d}z + \overline{a}(z, \overline{z})\,\mathrm{d}\overline{z}\right)^2 + c(z, \overline{z})^2\,\mathrm{d}z\mathrm{d}\overline{z}. \tag{2.7}$$

Crucially for our analysis, the authors in [20] showed that the deformations of the THM couple to Q-exact terms in the Lagrangian. Hence, given a fixed THF the free energy does not depend on the functions $h(\tau, z, \overline{z})$ and $c(\tau, z, \overline{z})$ in (2.4). Furthermore, [20] showed that the only deformations of the foliation that affect the free energy are those in the components $\xi^z$ and $\Phi^z{}_{\overline{z}}$. This means in particular that we can rescale $\xi$ without affecting the partition function. We now apply these results to the sphere and its deformations. Infinitesimal deformations of the THF away from the round sphere were already analyzed in [20].

## 2.1. The round sphere and the squashed sphere

In our first application we explain the squashing independence of the partition function of supersymmetric theories on the $SU(2) \times U(1)$ deformed sphere of [3]. For this we start by discussing supersymmetry on a round sphere of radius $r_3$. Using coordinates related to the Hopf fibration the metric takes the form

$$\mathrm{d}s^2_{\mathrm{round}} = \frac{r_3^2}{4}\left(\mathrm{d}\psi + \cos\theta\mathrm{d}\phi\right)^2 + \frac{r_3^2}{4}\left(\mathrm{d}\theta^2 + \sin^2\theta\mathrm{d}\phi^2\right). \tag{2.8}$$

For convenience we choose the left-invariant frame

$$\begin{aligned}
e^1 &= -\frac{r_3}{2}\left(\sin\psi\mathrm{d}\theta - \sin\theta\cos\psi\mathrm{d}\phi\right), \\
e^2 &= \frac{r_3}{2}\left(\cos\psi\mathrm{d}\theta + \sin\theta\sin\psi\mathrm{d}\phi\right), \\
e^3 &= -\frac{r_3}{2}\left(\mathrm{d}\psi + \cos\theta\mathrm{d}\phi\right),
\end{aligned} \tag{2.9}$$

in which all constant spinors $\zeta, \widetilde{\zeta}$ satisfy the Killing spinor equations in (2.1) for the background fields

$$A = V = 0, \qquad H = -\frac{\mathrm{i}}{2r_3}. \tag{2.10}$$

---

[4]To compute this contraction of spinors, the index on $\zeta_\alpha$ has to be raised to $\zeta^\alpha = \epsilon^{\alpha\beta}\zeta_\beta$ using the 2-index Levi-Civita tensor, $\epsilon_{12} = -\epsilon^{12} = 1$. Then the contraction reads $\zeta^\alpha (\gamma^\mu)_\alpha{}^\beta \widetilde{\zeta}_\beta$ for $(\gamma^\mu)_\alpha{}^\beta$ the Pauli-matrix with the frame index transformed into a coordinate index $\mu$ using the frame.

These spinors are invariant under the action of $SU(2)_{\mathrm{L}} \subset SO(4) \equiv SU(2)_{\mathrm{L}} \times SU(2)_R$, and the preserved supergroup $SU(2|1) \ltimes SU(2)_{\mathrm{L}}$. Out of the four supercharges preserved by the background let us focus on the pair of Killing spinors

$$\zeta = (1,0)^{\mathrm{T}}, \qquad \widetilde{\zeta} = (0,1)^{\mathrm{T}} \tag{2.11}$$

For these the Killing vector defined in (2.6) is given by

$$K = -\frac{2}{r_3}\partial_\psi = \xi \,, \tag{2.12}$$

where the second equality comes from the fact that $K$ is already normalized. We can then write the round sphere metric in the form (2.4) by using the adapted coordinates

$$\tau = -\frac{r_3}{2}\psi, \qquad z = \cot\frac{\theta}{2}e^{\mathrm{i}\phi}. \tag{2.13}$$

We then find that the functions in the transversaly hermitian metric are

$$h = -r_3\frac{\mathrm{i}}{4z}\frac{1-z\overline{z}}{1+z\overline{z}}, \qquad c = \frac{r_3}{1+z\overline{z}}. \tag{2.14}$$

We now compare this to the squashed sphere. Using the same coordinates as in the round case, deforming the metric on the sphere while preserving an $SU(2) \times U(1)$ isometry boils down to rescaling one of the coefficients. This leads to the metric

$$\mathrm{d}s^2_{\mathrm{squashed}} = \frac{r_3^2}{4}\left(\frac{b+b^{-1}}{2}\right)^2(\mathrm{d}\psi + \cos\theta\mathrm{d}\phi)^2 + \frac{r_3^2}{4}\left(\mathrm{d}\theta^2 + \sin^2\theta\mathrm{d}\phi^2\right). \tag{2.15}$$

We choose the frame to be proportional to the left-invariant frame of the round sphere, such that $e^3$ picks up the prefactor $(b+b^{-1})/2$ but the other components are unchanged. If we set the background fields to

$$H = -\frac{\mathrm{i}}{r_3}\frac{1}{b+b^{-1}}\left(1 - \left(\frac{b-b^{-1}}{2}\right)^2\right), \qquad V = -A = \frac{1}{8}\left(b-b^{-1}\right)^2(\mathrm{d}\psi + \cos\theta\mathrm{d}\phi)\,, \tag{2.16}$$

we preserve the pair of constant Killing spinors in (2.11) and the full supergroup is $SU(1|1) \ltimes SU(2)$. Thus, we preserve half the supercharges as compared to the round sphere. The Killing vector then takes the form

$$K = -\frac{4}{(b+b^{-1})r_3}\partial_\psi\,, \tag{2.17}$$

and we observe that the leaves of the foliation are preserved by the squashing. In terms of the adapted coordinates of the round sphere this means that $\xi^z$ still vanishes. Similarly one also finds that $\Phi^z_{\overline{z}}$ is unchanged by this squashing. Consequently the free energy is $b$-independent.

## 2.2. The ellipsoid and the squashed sphere

For our second case we continue with the squashed sphere metric in (2.15). By choosing a different set of background fields it is possible to preserve the supergroup $SU(2|1) \ltimes U(1)_{\mathrm{R}}$. In this subsection we show that the corresponding THF is equivalent to the one on the ellipsoid and hence the partition functions on the two spaces coincide.

Let us consider the squashed sphere with the same frame as before, but now with the background fields

$$H = \frac{\mathrm{i}}{4r_3} \left( b + b^{-1} \right), \qquad V = -A = \frac{1}{4} \left( b + b^{-1} \right) \left( b - b^{-1} \right) \left( \mathrm{d}\psi + \cos\theta \mathrm{d}\phi \right). \tag{2.18}$$

Defining $e^{\mathrm{i}\Theta} = -b$ and mapping the sphere into $SU(2)$ by setting

$$g = \begin{pmatrix} \cos\frac{\theta}{2} e^{\frac{\mathrm{i}}{2}(\phi+\psi)} & \sin\frac{\theta}{2} e^{\frac{\mathrm{i}}{2}(\phi-\psi)} \\ -\sin\frac{\theta}{2} e^{-\frac{\mathrm{i}}{2}(\phi-\psi)} & \cos\frac{\theta}{2} e^{\frac{-\mathrm{i}}{2}(\phi+\psi)} \end{pmatrix}, \tag{2.19}$$

one finds that for any choice of constant spinors $\zeta_0, \widetilde{\zeta}_0$, the spinors

$$\zeta = e^{\frac{\mathrm{i}}{2}\Theta\sigma_3} \cdot g^{-1} \cdot \zeta_0, \qquad \widetilde{\zeta} = e^{-\frac{\mathrm{i}}{2}\Theta\sigma_3} \cdot g^{-1} \cdot \widetilde{\zeta}_0 \tag{2.20}$$

satisfy the equation (2.1). Thus we have four independent conserved supercharges giving us an $SU(2|1) \ltimes U(1)$ supergroup.

To study the THF we choose the pair of Killing spinors,

$$\zeta_0 = \sqrt{b + b^{-1}} \, (1, 0)^{\mathrm{T}}, \qquad \widetilde{\zeta}_0 = \sqrt{b + b^{-1}} \, (0, -1)^{\mathrm{T}}, \tag{2.21}$$

which give the Killing vector

$$K = \zeta\gamma^\mu\widetilde{\zeta} = \frac{2}{r_3} \left( b - b^{-1} \right) \partial_\psi + \frac{2}{r_3} \left( b + b^{-1} \right) \partial_\phi. \tag{2.22}$$

To put $K$ into the form in (2.3) we define new coordinates

$$\widetilde{\psi} = \frac{r_3}{2} \left( \psi - \frac{b - b^{-1}}{b + b^{-1}} \phi \right), \qquad \tau = \frac{r_3}{2 \left( b + b^{-1} \right)} \phi, \tag{2.23}$$

such that $K = \partial_\tau$ and the metric takes the form

$$ds^2_{\text{squashed}} = \left(\frac{b + b^{-1}}{2}\right)^2 \left(b + b^{-1} + (b - b^{-1})\cos\theta\right)^2 \left(d\tau + \frac{b - b^{-1} + (b + b^{-1})\cos\theta}{(b + b^{-1} + (b - b^{-1})\cos\theta)^2}d\widetilde{\psi}\right)^2$$
$$+ \frac{r_3^2}{4}\left(d\theta^2 + \frac{4}{r_3^2 \left(b + b^{-1} + (b - b^{-1})\cos\theta\right)^2}\left(b + b^{-1}\right)^2 \sin^2\theta d\widetilde{\psi}^2\right).$$

$$(2.24)$$

We next consider the ellipsoid. Given some smooth function $f(\vartheta)$ satisfying $f(0) = \ell_2$ and $f(\frac{\pi}{2}) = \ell_1$, the ellipsoid metric is given by

$$ds^2_{\text{ellipsoid}} = f(\vartheta)^2 d\vartheta^2 + \ell_1^2 \cos^2\vartheta d\phi_1^2 + \ell_2^2 \sin^2\vartheta d\phi_2^2. \tag{2.25}$$

As was noted in [21], the index computation of [22] explains in a straightforward manner that the partition function on the ellipsoid does not depend on the form of $f$, but only on its values at $\vartheta = 0, \frac{\pi}{2}$. Thus we are free to choose

$$f(\vartheta) = \ell_1 \sin^2\vartheta + \ell_2 \cos^2\vartheta \tag{2.26}$$

which differs from the choice in [3, 22] but will facilitate comparison with the squashed sphere. In order to preserve supersymmetry on this metric background, we also have to turn on the background fields

$$H = \frac{i}{f}, \qquad A = -\frac{1}{2}\left(1 - \frac{\ell_1}{f}\right)d\phi_1 + \frac{1}{2}\left(1 - \frac{\ell_2}{f}\right)d\phi_2, \qquad V = 0. \tag{2.27}$$

The corresponding Killing spinors solving (2.1) then take the form

$$\zeta = e^{\frac{i}{2}(-\phi_2 + \phi_1)}\begin{pmatrix} e^{\frac{i}{2}\vartheta} \\ e^{-\frac{i}{2}\vartheta} \end{pmatrix}, \qquad \widetilde{\zeta} = e^{-\frac{i}{2}(-\phi_2 + \phi_1)}\begin{pmatrix} e^{\frac{i}{2}\vartheta} \\ -e^{-\frac{i}{2}\vartheta} \end{pmatrix}, \tag{2.28}$$

in the frame [3]

$$e^1 = \ell_1 \cos\vartheta d\phi_1, \quad e^2 = \ell_2 \sin\vartheta d\phi_2, \quad e^3 = f(\vartheta)d\vartheta. \tag{2.29}$$

This gives a $SU(1|1) \ltimes U(1)$ supergroup. In order to compare with the squashed sphere we make the change of variables

$$\psi = \phi_1 + \phi_2, \qquad \phi = \phi_1 - \phi_2, \qquad \theta = 2\vartheta. \tag{2.30}$$

The metric then takes the form

$$ds^2_{\text{ellipsoid}} = \frac{1}{4}f(\tfrac{\theta}{2})^2 d\theta^2 + \frac{1}{4}\left(\ell_1^2 \cos^2\frac{\theta}{2} + \ell_2^2 \sin^2\frac{\theta}{2}\right)(d\psi^2 + d\phi^2) + \frac{1}{2}\left(\ell_1^2 \cos^2\frac{\theta}{2} - \ell_2^2 \sin^2\frac{\theta}{2}\right)d\psi d\phi,$$

$$(2.31)$$

and the Killing vector generated by the Killing spinors in (2.28) is

$$K = 2\left(\frac{1}{\ell_1} - \frac{1}{\ell_2}\right)\partial_\psi + 2\left(\frac{1}{\ell_1} + \frac{1}{\ell_2}\right)\partial_\phi. \tag{2.32}$$

This is the same Killing vector as we found in (2.22) for the squashed sphere if we identify[5] $\ell_1 = r_3/b$ and $\ell_2 = r_3 b$. The normalized (w.r.t the respective metrics) vector fields are proportional to each other and the leaves of the corresponding foliations of $\mathbb{S}^3$ are the same.

Following the same procedure as for the squashed sphere we change to the coordinates $\left(\widetilde{\psi}, \theta, \tau\right)$ defined in (2.23). In these coordinates $K = \partial_\tau$ and the metric takes the form

$$ds^2_{\text{ellipsoid}} = 4\left(d\tau + \frac{\left(b + b^{-1}\right)\cos\theta - \left(b - b^{-1}\right)}{4}d\widetilde{\psi}\right)^2 + \frac{1}{4}f^2\left(d\theta^2 + f^{-2}\left(b + b^{-1}\right)^2\sin^2\theta d\widetilde{\psi}^2\right). \tag{2.33}$$

Introducing the adapted coordinates with $z = (1 + \cos\theta)^{\frac{r_3 b^{-1}}{2(b+b^{-1})}}(1 - \cos\theta)^{-\frac{r_3 b}{2(b+b^{-1})}}e^{i\widetilde{\psi}}$ for this metric it takes the form (2.4). In the same coordinates the squashed sphere metric (2.24) takes the form

$$ds^2 = (g(z, \overline{z})d\tau + h'(z, \overline{z})dz + \overline{h}'(z, \overline{z})d\overline{z})^2 + c'(z, \overline{z})^2 dz d\overline{z}. \tag{2.34}$$

From this we see that there is no consequential change in the THF when we go from the ellipsoid to the squashed sphere and the two partition functions are equal.

## 3. Symmetry enhancment from extended supergravity

In this section, by embedding the $\mathcal{N} = 2$ R-symmetry and the flavor symmetry in the $\mathcal{N} = 4$ R-symmetry we show that the special values of the mass parameter in $3d$ correspond to the existence of extra supersymmetries. The ideal framework for this computation is an off-shell formulation of $3d$ $\mathcal{N} = 4$ Poincaré supergravity but this is not readily available. We proceed by dimensionally reducing the 4d $\mathcal{N} = 2$ supergravity considered in [23,24]. The dimensional reduction gives three dimensional backgrounds to which $3d$ $\mathcal{N} = 4$ SYM naturally couple. Since ABJ(M) theory cannot be obtained this way our arguments do not apply directly. The correct setup to consider for ABJ(M) is the $\mathcal{N} = 6$ conformal supergravity [25] which we do in appendix C and show that the same conclusions hold.

The details of the dimensional reduction are provided in appendix B and we only state the main results here. The bosonic fields of the $3d$ backgrounds are

$$g_{\mu\nu} \qquad V_\mu \qquad A_\mu^{\text{R}_0} \qquad A_\mu^{\text{F}} \qquad H \qquad \sigma. \tag{3.1}$$

---

[5]With this identification $f = \frac{r_3}{2}\left(b + b^{-1} + \left(b - b^{-1}\right)\cos\theta\right)$.

Here $V_\mu$ is the dual graviphoton field strength which appears in $3d$ via the reduction of the four-dimensional metric. The four-dimensional $\mathcal{N} = 2$ superconformal algebra has $U(1)_{\rm r} \times SU(2)_{\rm R}$ R-symmetry. The gauge fields and the scalars are the dimensional reduction of the gauge field corresponding to the R-symmetry generated by the $U(1)_{\rm r} \times U(1)_{\rm R}$ subgroup[6]. From the $3d$ $\mathcal{N} = 2$ perspective, $A_\mu^{\rm R_0}$ corresponds to the gauge field for the R-symmetry while $A_\mu^{\rm F}$ and $\sigma$ are bosonic fields of a background $U(1)$ flavor multiplet.

Setting the variation of the gravitino to zero leads to the following generalized Killing spinor equations (GKSE)

$$
\left( \nabla_\mu - {\rm i} \left( A_\mu^{\rm R_0} + \frac{1}{2} V_\mu \right) - H \gamma_\mu + \varepsilon_{\mu\nu\rho} V^\nu \gamma^\rho \right) \xi^1 = 0
$$
$$
\left( \nabla_\mu + {\rm i} \left( A_\mu^{\rm R_0} - A_\mu^{\rm F} - \frac{1}{2} V_\mu \right) - (\sigma - H) \gamma_\mu + \varepsilon_{\mu\nu\rho} V^\nu \gamma^\rho \right) \xi^2 = 0 \,.
$$
(3.2)

The first equation is precisely the GKSE (2.1) for $3d$ $\mathcal{N} = 2$ supersymmetric backgrounds[7]. Requiring the variation of the dilatino to vanish gives

$$
\delta\chi^1 = (D + \sigma H) \xi^1 - \frac{1}{3} \left( \varepsilon^{\mu\nu\rho} \gamma_\rho \xi^1 \partial_\mu \left( A_\nu^{\rm R_0} - \frac{3}{2} A_\nu^{\rm F} \right) + \gamma^\mu \xi^1 \left( \partial_\mu + 2{\rm i} V_\mu \right) \left( H - \frac{3}{2}\sigma \right) \right) = 0,
$$
$$
\delta\chi^2 = (D + \sigma H) \xi^2 + \frac{1}{3} \left( \varepsilon^{\mu\nu\rho} \gamma_\rho \xi^2 \partial_\mu \left( A_\nu^{\rm R_0} + \frac{1}{2} A_\nu^{\rm F} \right) + \gamma^\mu \xi^2 \left( \partial_\mu + 2{\rm i} V_\mu \right) \left( H + \frac{1}{2}\sigma \right) \right) = 0.
$$
(3.3)

If $\xi^2 = 0$ then the constraints in (3.3) reproduce the conditions for a $3d$ $\mathcal{N} = 2$ supersymmetric theory coupled to a background $U(1)$ flavor multiplet. $\xi^1$ satisfies the Killing spinor equation for the background fields

$$
A^{\rm R_0} = -\frac{1}{2} V = -\frac{\left( b + b^{-1} \right) \left( b - b^{-1} \right)}{16} \left( {\rm d}\psi + \cos\theta {\rm d}\phi \right), \qquad H = {\rm i}\frac{b + b^{-1}}{4r_3}.
$$
(3.4)

It then follows that

$$
\varepsilon^{\mu\nu\rho} \partial_\mu A_\nu^{\rm R_0} \gamma_\rho + 2{\rm i} H V^\mu \gamma_\mu = 0
$$
(3.5)

and the constraints in (3.3) simplify considerably. For constant $\sigma$ and without imposing any further restrictions on $\xi^{1,2}$, (3.3) is satisfied for

$$
D = -\sigma H, \qquad A^{\rm F} = \frac{2{\rm i}\sigma r_3}{b + b^{-1}} V.
$$
(3.6)

For $\sigma = 2H$, the second GKSE is also satisfied for all $\xi^2$ of the same form (2.20) as $\xi^1$, hence

---

[6]We only allow the $SU(2)_{\rm R}$ symmetry gauge field to take values in the Cartan subalgebra. It would be interesting to study the generic dimensional reduction with arbitrary $SU(2)_{\rm R}$ gauge field as well as other fields in the $\mathcal{N} = 2$ Weyl multiplet but we do not attempt it here.

[7]We scale the $V_\mu$ in (2.1) and then identify $A_\mu = A_\mu^{\rm R_0} - \frac{3}{2} V_\mu$

the amount of supersymmetry is doubled for this choice of mass. For $\sigma = \mathrm{i}\frac{b-b^{-1}}{2r_3}$ and $\xi^2 = (1,0)^{\mathrm{T}}$ the second GKSE becomes

$$\nabla_\mu \xi^2 + \mathrm{i}\left(\frac{b-b^{-1}}{b+b^{-1}}\right) V_\mu \xi^2 + \frac{\mathrm{i}}{4}\frac{b+b^{-1}}{r_3}\gamma_\mu \xi^2 = 0, \tag{3.7}$$

which is indeed satisfied. $\xi^2 = (0,1)^{\mathrm{T}}$ solves the GKSE for $\sigma = -\mathrm{i}\frac{b-b^{-1}}{2r_3}$. So, in this case we see that we have two more supercharges. Note that these are exactly the supercharges preserved in the familiar squashing discussed in subsection 2.1.

## 4. $4d$ indices, $3d$ partition functions and symmetry enhancement

In this section we study the relation between partition functions on the three-dimensional squashed sphere and dimensional reduction of the four-dimensional indices [26–29]. We first derive relations with the $\mathcal{N}=1$ and $\mathcal{N}=2$ indices. We then show how various special values of the mass parameter in the $3d$ theory correspond to limits of the index where only states annihilated by extra supercharges contribute.

Consider a theory with $\mathcal{N}=1$ superconformal symmetry placed on $\mathbb{S}^1 \times \mathbb{S}^3$, with radii $r_1$ and $r_3$, respectively. The $\mathcal{N}=1$ superconformal group $SU(2,2|1)$ has bosonic generators $P_\mu, j_1, j_2, j_1^\pm, j_2^\pm$ for the $4d$ Poincaré algebra, $E$ and $K_\mu$ for dilations and special conformal transformations, and $\widetilde{r}$ for the $U(1)_R$ symmetry. Its fermionic generators are the supercharges $Q_\alpha, \widetilde{Q}_{\dot\alpha}$ and the conformal supercharges $S_\alpha, \widetilde{S}_{\dot\alpha}$ for $\alpha, \dot\alpha = +, -$.

We can build an index using the supercharge $\mathcal{Q} \equiv Q_-$ and its adjoint $\mathcal{Q}^\dagger \equiv S_+$. These satisfy the anti-commutation relations

$$\{\mathcal{Q}, \mathcal{Q}^\dagger\} = E - 2j_1 - \frac{3}{2}\widetilde{r}, \tag{4.1}$$

where the right hand side commutes with $\mathcal{Q}$ and $\mathcal{Q}^\dagger$. Since $\mathcal{Q}$ has R-charge $\widetilde{r}=1$, the combination $E - \frac{1}{2}\widetilde{r}$ also commutes with $\mathcal{Q}$ and $\mathcal{Q}^\dagger$. Hence, treating the $\mathbb{S}^1$ as Euclidean time we can construct the index

$$\mathrm{Tr}\,(-1)^F\, e^{-2\pi\frac{r_1}{r_3}(E-\frac{1}{2}\widetilde{r})} = \mathrm{Tr}(-1)^F\,(pq)^{\frac{1}{2}(E-\frac{1}{2}\widetilde{r})} \equiv \mathcal{I}(p,q), \tag{4.2}$$

where the trace is over the states in the Hilbert space on $\mathbb{S}^3$, and the fugacities are defined as

$$p = e^{-2\pi b v \frac{r_1}{r_3}} \qquad q = e^{-2\pi b^{-1} v \frac{r_1}{r_3}}, \tag{4.3}$$

with $v = 2(b+b^{-1})^{-1}$,

The index $\mathcal{I}(p,q)$ is equal to the path integral

$$\mathcal{I}(p,q) = \int D\phi\, e^{-S_{\mathbb{S}^1 \times \mathbb{S}^3} + \pi\frac{r_1}{r_3}\int_{\mathbb{S}^3} j_{\widetilde{r}}^0}, \tag{4.4}$$

where the path integration over the fields has periodic boundary conditions on $\mathbb{S}^1$ for both fermions and bosons. $j_{\tilde{r}}^0$ is the $U(1)_R$ charge density, while under dimensional reduction we have that

$$S_{\mathbb{S}^1 \times \mathbb{S}^3} = 2\pi r_1 S_{\mathbb{S}^3} \tag{4.5}$$

where $\mathbb{S}^3$ is the dimensionally reduced action on $\mathbb{S}^3$. Hence, we see that the combination

$$\widetilde{S}_{\mathbb{S}^3} = S_{\mathbb{S}^3} - \frac{1}{2}\int_{\mathbb{S}^3} j_{\tilde{r}}^0 \tag{4.6}$$

is the three-dimensional action that is invariant under supersymmetry transformations generated by $\mathcal{Q}$ and $\mathcal{Q}^\dagger$. Since $Q_+$ and $S_-$ also commute with $E - \frac{1}{2}\tilde{r}$, $\widetilde{S}_{\mathbb{S}^3}$ is invariant under these supersymmetry transformations as well. Hence, the theory on the round sphere generically preserves four supersymmetries.

If the theory also has some flavor symmetries, labeled by $F_i$, then these too can be inserted into the index since all $F_i$ commute with $\mathcal{Q}$. These are put in by turning on some appropriate background gauge fields along $\mathbb{S}^1$. Each $F_i$ comes with a fugacity $t_i$, which are given by

$$t_i = e^{-2\pi i r_1 m_i} \tag{4.7}$$

where $m_i$ is the real mass parameter that appears in the dimensionally reduced $3d$ theory. It is related to the dimensionless parameter $\mu_i$ that appears in the one-loop determinants by

$$m_i = \frac{\upsilon \mu_i}{r_3}. \tag{4.8}$$

The index then takes the form

$$\mathcal{I}(p, q, t_i) = \text{Tr}(-1)^F (pq)^{\frac{1}{2}(E - \frac{1}{2}\tilde{r})} \prod_i t_i^{F_i}. \tag{4.9}$$

We will eventually be interested in a flavor symmetry that descends from $\mathcal{N} = 2$ superconformal symmetry in four dimensions. To this end we extend the supercharges to $Q_{a\alpha}, \widetilde{Q}_{a\dot{\alpha}}, S_{a\alpha}, \widetilde{S}_{a\dot{\alpha}}$, where $a = 1, 2$, and we identify these with the $\mathcal{N} = 1$ supercharges as $Q_\alpha = Q_{1\alpha}$, $S_\alpha = S_{1\alpha}$, $\widetilde{Q}_{\dot{\alpha}} = \widetilde{Q}_{2\dot{\alpha}}$, $\widetilde{S}_{\dot{\alpha}} = \widetilde{S}_{2\dot{\alpha}}$. The $\mathcal{N} = 2$ R-symmetry is $SU(2) \times U(1)$ with generators R, $R^\pm$ and r. Comparing the anti-commutation relations, we see that the $\mathcal{N} = 1$ $U(1)_R$ generator is related to the $\mathcal{N} = 2$ $SU(2) \times U(1)$ generators as $\tilde{r} = \frac{2}{3}(r + 2R)$. The $\mathcal{N} = 2$ supersymmetry generators that are not part of the $\mathcal{N} = 1$ generators can have flavor charges assigned to them which we normalize to $\pm 1$. Hence, we have a $U(1)$ flavor symmetry in the $\mathcal{N} = 1$ theory, given by $f = r - R$.

Since $E - \frac{1}{2}\tilde{r}$ does not commute with the extra supersymmetries, the $3d$ action $\widetilde{S}_{\mathbb{S}^3}$ is not invariant under their transformations. However, we can improve this by using the flavor symmetry. In particular, if we instead consider the combination $E - \frac{1}{2}\tilde{r} + \frac{1}{3}f = E - R$, then this also commutes

| $\mathcal{Q}$ | $SU(2)_1$ | $SU(2)_2$ | $SU(2)_R$ | $U(1)_r$ | $r_0$ | $f$ |
|---|---|---|---|---|---|---|
| $\mathcal{Q}_{1-}$ | $-\frac{1}{2}$ | $0$ | $\frac{1}{2}$ | $\frac{1}{2}$ | $1$ | $0$ |
| $\mathcal{Q}_{1+}$ | $\frac{1}{2}$ | $0$ | $\frac{1}{2}$ | $\frac{1}{2}$ | $1$ | $0$ |
| $\mathcal{Q}_{2-}$ | $-\frac{1}{2}$ | $0$ | $-\frac{1}{2}$ | $\frac{1}{2}$ | $-1$ | $1$ |
| $\mathcal{Q}_{2+}$ | $\frac{1}{2}$ | $0$ | $-\frac{1}{2}$ | $\frac{1}{2}$ | $-1$ | $1$ |
| $\widetilde{\mathcal{Q}}_{1\dot{-}}$ | $0$ | $-\frac{1}{2}$ | $\frac{1}{2}$ | $-\frac{1}{2}$ | $1$ | $-1$ |
| $\widetilde{\mathcal{Q}}_{1\dot{+}}$ | $0$ | $\frac{1}{2}$ | $\frac{1}{2}$ | $-\frac{1}{2}$ | $1$ | $-1$ |
| $\widetilde{\mathcal{Q}}_{2\dot{-}}$ | $0$ | $-\frac{1}{2}$ | $-\frac{1}{2}$ | $-\frac{1}{2}$ | $-1$ | $0$ |
| $\widetilde{\mathcal{Q}}_{2\dot{+}}$ | $0$ | $\frac{1}{2}$ | $-\frac{1}{2}$ | $-\frac{1}{2}$ | $-1$ | $0$ |

Table 1: For each supercharge $\mathcal{Q}$ we list its quantum numbers. Here $a = 1, 2$ are the $SU(2)_R$ indices and $\alpha = \pm$, $\dot{\alpha} = \pm$ are the Lorentz indices. $E$ is the conformal dimension, $(j_1, j_2)$ the Cartan generators of the $SU(2)_1 \otimes SU(2)_2$ isometry group, $(R, r)$, the Cartan generators of the $SU(2)_R \otimes U(1)_r$ R-symmetry group, $r_0$ is the $3d$ R-charge and $f$ is the flavor charge.

with $\widetilde{Q}_{1\dot{\alpha}}$ and $\widetilde{S}_{1\dot{\alpha}}$. Hence turning on a Wilson line along the $\mathbb{S}^1$ coupled to the appropriate charges enhances the number of supersymmetries on $\mathbb{S}^3$ to eight. The flavor symmetry also shifts the $3d$ $U(1)$ to $r_0 = \widetilde{r} - \frac{2}{3}f = 2R$. With the shift $Q_{1\alpha}$ and $\widetilde{Q}_{1\dot{\alpha}}$ both have $r_0 = 1$ while $S_{2\alpha}$ and $\widetilde{S}_{2\dot{\alpha}}$ both have $r_0 = -1$. A summary of the various charges for the different supercharges are shown in table 1[8].

Next consider squashing the $\mathbb{S}^3$ such that it preserves an $SU(2) \times U(1)$ isometry. The charge $j_2$ rotates the angle $\psi$ which parameterizes the Hopf fiber over the $\mathbb{S}^2$. We can then obtain the $SU(2) \times U(1)$ squashed metric by twisted dimensional reduction, where the rotation on the fiber as one circles the $\mathbb{S}^1$ is given by

$$\left(x^4, \psi\right) \sim \left(x^4 + 2\pi r_1, \psi + 4\pi i \frac{b - b^{-1}}{b + b^{-1}} \frac{r_1}{r_3}\right). \tag{4.10}$$

The effect of the twisting is to insert the factor

$$\exp\left(-4\pi \frac{b - b^{-1}}{b + b^{-1}} \frac{r_1}{r_3} j_2\right) = \left(\frac{p}{q}\right)^{j_2} \tag{4.11}$$

into the index, giving us

$$\mathcal{I}_{SU(2) \times U(1)}(p, q) = \text{Tr}(-1)^F p^{\frac{1}{2}(E - \frac{1}{2}\widetilde{r} + \frac{1}{3}f + 2j_2)} q^{\frac{1}{2}(E - \frac{1}{2}\widetilde{r} + \frac{1}{3}f - 2j_2)}. \tag{4.12}$$

This index is well-defined since $j_2$ commutes with $\mathcal{Q}$. It also preserves the supersymmetry generated by $Q_{1+}$, but breaks the $\widetilde{Q}_{1\dot{\alpha}}$ supersymmetries. Hence, four supersymmetries are preserved in general. In fact, without breaking any further supersymmetries we can combine this with a flavor term to

---

[8]The table is an amended version of table 1 in [30].

give

$$\mathcal{I}_{SU(2)\times U(1)}(p,q,t) = \text{Tr}(-1)^F p^{\frac{1}{2}(E-\frac{1}{2}\widetilde{r}+\frac{1}{3}f+2j_2)} q^{\frac{1}{2}(E-\frac{1}{2}\widetilde{r}+\frac{1}{3}f-2j_2)} t^f \tag{4.13}$$

where $t = e^{-2\pi i r_1 m}$ and $m$ has the same relation to the dimensionless parameter $\mu$ as in (4.8). This last insertion adds a mass-term to the $3d$ action.

We now show that for special values of $\mu$ the index simplifies. For

$$\mu = -\frac{i}{2}(b-b^{-1})\,, \tag{4.14}$$

the index becomes

$$\mathcal{I}_{SU(2)\times U(1)}(p,q) = \text{Tr}(-1)^F p^{\frac{1}{2}(E-\frac{1}{2}\widetilde{r}+\frac{1}{3}f+2j_2+f)} q^{\frac{1}{2}(E-\frac{1}{2}\widetilde{r}+\frac{1}{3}f-2j_2-f)}\,. \tag{4.15}$$

Examining the charges in table 1, we see that $\widetilde{Q}_{1\dot{+}}$ commutes with $2j_2 + f$. Hence, with this choice of mass two extra supersymmetries are preserved, giving six altogether. Moreover, the exponent of $p$ equals $\{\widetilde{Q}_{1\dot{+}}, \widetilde{Q}^{\dagger}_{1\dot{+}}\}$ so we can also rewrite the index in the form

$$\mathcal{I}_{SU(2)\times U(1)}(p,q) = \text{Tr}(-1)^F q^{\frac{1}{2}(E-\frac{1}{2}\widetilde{r}+\frac{1}{3}f-2j_2-f)}. \tag{4.16}$$

This makes the squashing independence explicit as all $b$ dependence can be absorbed into a rescaling of the $\mathbb{S}^3$ radius $r_3$. For $\mu = i\frac{b-b^{-1}}{2}$ supersymmetries associated with $\widetilde{Q}_{1\dot{-}}$ are preserved. This is consistent with the analysis in section 3.

The other point we are interested in is

$$\mu = i\frac{b+b^{-1}}{2} \tag{4.17}$$

where the index takes the form

$$\mathcal{I}_{SU(2)\times U(1)}(p,q) = \text{Tr}(-1)^F p^{\frac{1}{2}(E-\frac{1}{2}\widetilde{r}+\frac{1}{3}f+2j_2-f)} q^{\frac{1}{2}(E-\frac{1}{2}\widetilde{r}+\frac{1}{3}f-2j_2-f)} \tag{4.18}$$

$$= \text{Tr}(-1)^F (pq)^{\frac{1}{2}(E-r)} \left(\frac{p}{q}\right)^{j_2}. \tag{4.19}$$

The exponents in this index commute with all supercharges without tildes, giving a total of eight preserved charges. Contributions to the index then come from multiplets satisfying $E = r$, meaning that we can simplify the index to

$$\mathcal{I}_{SU(2)\times U(1)}(p,q) = \text{Tr}(-1)^F \left(\frac{p}{q}\right)^{j_2}. \tag{4.20}$$

From here we see the squashing independence of the partition function as again $b$ can be absorbed by rescaling $r_3$. Note that enhancement at $\mu = -i\frac{b+b^{-1}}{2}$ happens if one exchanges the roles of the

two Lorentz $SU(2)$'s.

In a similar fashion we can construct an index that reduces to the familiar squashing case [3]. Taking the twisted compactification from before, we would like the index to be of the form

$$\text{Tr}(-1)^F e^{-2\pi \frac{r_1}{r_3}(E+2iuj_2+\mathcal{O})} \tag{4.21}$$

where we have defined $u = -i\frac{b-b^{-1}}{b+b^{-1}}$. The operator $\mathcal{O}$ is chosen such that the index will be with respect to the supercharge $\widetilde{\mathcal{Q}}_{2\dot{-}}$. Given that this supercharge commutes with $E - 2j_2 + 2R + r$ and $E + j_2$, it is easy to see that $\mathcal{O} = \frac{1}{3}(1 - 2iu)(2R + r)$. Then writing again an index in terms of $p$ and $q$ we find

$$\mathcal{I}_{\text{HH}} = \text{Tr}(-1)^F p^{\frac{1}{2}(E+2j_2-\frac{2}{3}R-\frac{1}{3}r)} q^{\frac{1}{2}(E-2j_2+2R+r)} \tag{4.22}$$

$$= \text{Tr}(-1)^F p^{\frac{1}{2}(E+2j_2-\frac{2}{3}R-\frac{1}{3}r)}. \tag{4.23}$$

In going to the second line we used that the exponent of $q$ is equal to $\{\widetilde{\mathcal{Q}}_{2\dot{-}}, \widetilde{\mathcal{Q}}_{2\dot{-}}^\dagger\}$. The last line shows that the dependence on $b$ of this index is trivial in the sense that it can be absorbed into a rescaling of the $\mathbb{S}^3$ radius $r_3$. After dimensional reduction this translates into the squashing independence we discussed in section 2.1.

## Acknowledgements

We thank Shai Chester and Guido Festuccia for helpful comments on the manuscript. This research is supported in part by Vetenskapsrådet under grants #2016-03503, #2018-05572 and #2020-03339, and by the Knut and Alice Wallenberg Foundation under grant Dnr KAW 2015.0083.

## A. Partition functions of $3d$ $\mathcal{N} = 2$ theories

Here we summarize the contributions of different multiplets to the partition function and resulting simplifications for special values of mass parameters.

The general form of the partition function is

$$\mathcal{Z} = \int \mathrm{d}^{r_G}\sigma\, e^{-S_0} \left( \prod_{\alpha\in\Delta_+} \langle\alpha,\sigma\rangle^2 \right) Z_{1-\text{loop}}(b,\sigma), \tag{A.1}$$

where $r_G$ is the rank of the gauge group, $S_0$ is the action evaluated at the localization locus. Only the Chern-Simons terms and the FI terms contribute to $S_0$. The product is over the positive roots of the Lie algebra. $Z_{1-\text{loop}}$ is the one-loop determinant which is a product of one-loop determinants for an $\mathcal{N} = 2$ vector multiplet and various chiral multiplets depending on the matter content of the

theory. The basic one-loop determinants are for the vector multiplet

$$Z^{\text{vec}}_{\mathcal{N}=2} = \prod_{\alpha \in \Delta_+} \frac{\sinh\left(\pi b \langle \sigma, \alpha \rangle\right) \sinh\left(\pi b^{-1} \langle \sigma, \alpha \rangle\right)}{\langle \sigma, \alpha \rangle^2}, \tag{A.2}$$

and for an $\mathcal{N} = 2$ chiral multiplet with R-charge $q$, flavor charge $F$ and mass $\mu$ in any representation of the gauge group

$$Z^{\text{ch}}_{\mathcal{N}=2}(q, F, \mu) = \prod_{\rho \in R} s_b\left(\frac{iQ}{2}(1-q) + F\mu - \langle \sigma, \rho \rangle\right), \tag{A.3}$$

where $s_b(x)$ is the double sine function, defined as the (regularized) infinite product

$$s_b(x) = \prod_{m,n \geq 0} \frac{(m + \frac{1}{2})b + (n + \frac{1}{2})b^{-1} - ix}{(m + \frac{1}{2})b + (n + \frac{1}{2})b^{-1} + ix}. \tag{A.4}$$

Using these formulae it is easy to confirm that for a chiral in the adjoint representation

$$
\begin{aligned}
Z^{\text{ch}}_{\mathcal{N}=2}\left(\frac{1}{2}, \frac{1}{2}, i\frac{b+b^{-1}}{2}\right) &= \prod_{\alpha \in \Delta_+} \frac{1}{\sinh\left(\pi b \langle \sigma, \alpha \rangle\right) \sinh\left(\pi b^{-1} \langle \sigma, \alpha \rangle\right)}, \\
Z^{\text{ch}}_{\mathcal{N}=2}\left(\frac{1}{2}, \frac{1}{2}, -i\frac{b+b^{-1}}{2}\right) &= Z^{\text{ch}}_{\mathcal{N}=2}(1, F, 0) = 1, \\
Z^{\text{ch}}_{\mathcal{N}=2}\left(1, 1, \pm i\frac{b+b^{-1}}{2}\right) &= \prod_{\alpha \in \Delta_+} \left(\sinh\left(\pi b \langle \sigma, \alpha \rangle\right) \sinh\left(\pi b^{-1} \langle \sigma, \alpha \rangle\right)\right)^{\mp 1}, \\
Z^{\text{ch}}_{\mathcal{N}=2}\left(1, 1, \pm i\frac{b-b^{-1}}{2}\right) &= b^{\mp \text{rank } G} \prod_{\alpha \in \Delta_+} \left(\frac{\sinh \pi b \langle \sigma, \alpha \rangle}{\sinh \pi b^{-1} \langle \sigma, \alpha \rangle}\right)^{\mp 1}.
\end{aligned}
\tag{A.5}
$$

For $\mathcal{N} = 4$ multiplets, the R-charge $q$ of constituent chirals is the sum of the charges under $U(1) \times U(1) \subset SU(2) \times SU(2) = SO(4)_R$ while their flavor charge is the difference. This means that twisted multiplets have the opposite flavor charge and we will not need to list them separately in what follows. The $\mathcal{N} = 4$ vector multiplet consists of an $\mathcal{N} = 2$ vector multiplet and an appropriate adjoint $\mathcal{N} = 2$ chiral multiplet with R-charge $q = 1$ and flavor charge 1. The corresponding one-loop determinant has the form

$$Z^{\text{vec}}_{\mathcal{N}=4} = Z^{\text{vec}}_{\mathcal{N}=2} Z^{\text{ch}}_{\mathcal{N}=2}(1, 1, \mu) \tag{A.6}$$

$$= \begin{cases} b^{\mp \text{rank } G} \prod_{\alpha \in \Delta_+} \frac{\sin\left(\pi b^{\mp 1} \langle \sigma, \alpha \rangle\right)^2}{\langle \sigma, \alpha \rangle^2}, & \mu = \pm i\frac{b-b^{-1}}{2}, \\ \prod_{\alpha \in \Delta_+} \frac{1}{\langle \sigma, \alpha \rangle^2}, & \mu = +i\frac{b+b^{-1}}{2}. \end{cases} \tag{A.7}$$

Note that for the $\mu = \pm i\frac{b-b^{-1}}{2}$ case the above expression together with the integration measure on the Cartan algebra of the gauge group and the Vandermonde determinant only depend on $b^{\mp 1}\sigma$, making the squashing independence explicit. This rescaling of the Coulomb branch parameter

matches with the rescaling we found for the index 4.16, confirming the choice of sign that $\mu$ comes with in the above expression A.6.

The $\mathcal{N}=4$ hypermultiplet consists of two $\mathcal{N}=2$ chiral multiplet with R-charge $\frac{1}{2}$ and opposite gauge charges. Besides the mass $\mu$ that we tune, the hypermultiplet also admits a real mass $m$ that preserves $\mathcal{N}=4$. So the one-loop determinant in a representation $\mathcal{R}$ has the form

$$Z_{\mathcal{N}=4}^{\text{hyp}}(\mu,m) = Z_{\mathcal{N}=2}^{\text{ch},\mathcal{R}}(\frac{1}{2},-\frac{1}{2},\mu+2m)Z_{\mathcal{N}=2}^{\text{ch},\overline{\mathcal{R}}}(\frac{1}{2},-\frac{1}{2},\mu-2m) \tag{A.8}$$

$$= \begin{cases} \displaystyle\prod_{\rho\in\mathcal{R}}\frac{1}{\cosh(\pi b^{\mp 1}(\langle\sigma,\rho\rangle+m))}, & \mu=\pm\mathrm{i}\dfrac{b-b^{-1}}{2} \\ 1, & \mu=+\mathrm{i}\dfrac{b+b^{-1}}{2} \end{cases} \tag{A.9}$$

Note that once again the first case only depends on $b^{\mp 1}\sigma$. With these ingredients it is now easy to put the various pieces together and obtain squashing independent partition functions as well as partition functions with very simple squashing dependence. An example illustrating this squashing independence is the $N_f$-model in appendix A.2.

## A.1. Another squashing independent point for ABJ(M)

In this section of the appendix we show squashing independence for the mass deformed ABJ(M) theory at a special mass given below. The partition function of mass-deformed ABJ(M) theory on the squashed three-sphere is given by

$$Z(b;m_1,m_2,m_3) = e^{\frac{\mathrm{i}\pi}{12k}(b-b^{-1})M(M^2-1)}\int\frac{d^{N+M}\mu d^N\nu}{N!(N+M)!}e^{\mathrm{i}\pi k(\sum_i\nu_i^2-\sum_a\mu_a^2)}$$
$$\times\prod_{a>b}4\sinh\left(\pi b(\mu_a-\mu_b)\right)\sinh\left(\pi b^{-1}(\mu_a-\mu_b)\right)\prod_{i>j}4\sinh\left(\pi b(\nu_i-\nu_j)\right)\sinh\left(\pi b^{-1}(\nu_i-\nu_j)\right)$$
$$\times\prod_{i,a}\left\{s_b\left[\frac{\mathrm{i}Q}{4}-\left(\mu_a-\nu_i+\frac{m_1+m_2+m_3}{2}\right)\right]s_b\left[\frac{\mathrm{i}Q}{4}-\left(\mu_a-\nu_i+\frac{m_1-m_2-m_3}{2}\right)\right]\right. \tag{A.10}$$
$$\left.\times s_b\left[\frac{\mathrm{i}Q}{4}-\left(-\mu_a+\nu_i+\frac{-m_1-m_2+m_3}{2}\right)\right]s_b\left[\frac{\mathrm{i}Q}{4}-\left(-\mu_a+\nu_i+\frac{-m_1+m_2-m_3}{2}\right)\right]\right\}$$

In the following we will show that this is squashing independent for $m_3=\mathrm{i}\frac{Q}{2}$ and $m_1=m_2$. Thus we will look at $\mathcal{Z}=Z(b;\frac{m_+-m_-}{2},\frac{m_++m_-}{2},\mathrm{i}\frac{Q}{2})$ where $m_\pm=m_1\pm m_2$.

With the definition eq. (A.4) and the identity

$$\sinh(z) = z\prod_{k\geq 1}\left(1+\frac{z^2}{\pi^2 k^2}\right), \tag{A.11}$$

it is easy to see that, after rescaling $(\mu,\nu)\to(\mu,\nu)/(2\pi)$ and putting some (divergent) constant

factors into $\mathcal{N}_1$, our partition function of interest simplifies to

$$\mathcal{Z} = \mathcal{N}_1 e^{\frac{i\pi}{12k}(b-b^{-1})M(M^2-1)}$$

$$\times \int d^{N+M}\mu d^N\nu\, e^{i\pi k(\sum_i \nu_i^2 - \sum_a \mu_a^2)} \prod_{a>b} 4\sinh\left(b\frac{\mu_a - \mu_b}{2}\right)\sinh\left(b^{-1}\frac{\mu_a - \mu_b}{2}\right)$$

$$\times \prod_{i>j} 4\sinh\left(b\frac{\nu_i - \nu_j}{2}\right)\sinh\left(b^{-1}\frac{\nu_i - \nu_j}{2}\right)\prod_{i,a}\frac{1}{\sinh\left(b^{-1}\frac{\mu_a - \nu_i - \pi m_-}{2}\right)\sinh\left(b\frac{\mu_a - \nu_i - \pi m_-}{2}\right)}.$$

$$(A.12)$$

Applying the Cauchy determinant formula, this can be rewritten as

$$\mathcal{Z} = \mathcal{N}_1 e^{\frac{i\pi}{12k}(b-b^{-1})M(M^2-1)}\int d^{N+M}\mu d^N\nu\; e^{\frac{ik}{4\pi}\left(\sum_i \nu_i^2 - \sum_a \mu_a^2\right) - \frac{M}{2}\sum_r(Q\mu_r - \pi Q m_-) + \frac{Q}{2}\sum_s Q\nu_s}$$

$$\times \det\left(\theta_{N,l}\frac{1}{2\sinh\frac{b^{-1}(\mu_j - \nu_l - \pi m_-)}{2}} + \theta_{l,N+1}e^{(M+N-l+\frac{1}{2})b^{-1}(\mu_j - \pi m_-)}\right)$$

$$\times \det\left(\theta_{N,l}\frac{1}{2\sinh\frac{b(\mu_j - \nu_l - \pi m_-)}{2}} + \theta_{l,N+1}e^{(M+N-l+\frac{1}{2})b(\mu_j - \pi m_-)}\right).$$

$$(A.13)$$

By expanding the determinants and reordering integration variables to get rid of one summation, this can be simplified to

$$\mathcal{Z} = \mathcal{N}_2 \sum_{\sigma \in S_{M+N}} (-1)^\sigma \int d^{N+M}\mu d^N\nu\; e^{\frac{ik}{4\pi}\left(\sum_i \nu_i^2 - \sum_a \mu_a^2\right) - \frac{M}{2}\sum_r(Q\mu_r - \pi Q m_-) + \frac{Q}{2}\sum_s Q\nu_s + \frac{i\pi}{12k}(b-b^{-1})M(M^2-1)}$$

$$\times \frac{\prod_{j=N+1}^{N+M} e^{(M+N-l+\frac{1}{2})b^{-1}(\mu_j - \pi m_-)}e^{(M+N-l+\frac{1}{2})b(\mu_{\sigma(j)} - \pi m_-)}}{\prod_{j=1}^N 4\sinh\frac{b^{-1}(\mu_j - \nu_j - \pi m_-)}{2}\sinh\frac{b(\mu_{\sigma(j)} - \nu_j - \pi m_-)}{2}},$$

$$(A.14)$$

where we put some more irrelevant factors into the new normalization factor $\mathcal{N}_2$. Fourier transforming the hyperbolic functions we get

$$\mathcal{Z} = \mathcal{N}_3 \int d^{N+M}\mu d^N\nu d^N p\, d^{M+N}q\; e^{\frac{ik}{4\pi}\left(\sum_i \nu_i^2 - \sum_a \mu_a^2\right) - \frac{M}{2}\sum_r(Q\mu_r - \pi Q m_-) + \frac{Q}{2}\sum_s Q\nu_s + \frac{i\pi}{12k}(b-b^{-1})M(M^2-1)}$$

$$\times \prod_{j=1}^N \tanh\left(\frac{p_j}{2}\right)e^{\frac{ip_j}{2\pi b}(\mu_j - \nu_j - \pi m_-)}\prod_{j=N+1}^{N+M} e^{(M+N-l+\frac{1}{2})b^{-1}(\mu_j - \pi m_-)}\prod_{s=1}^N \tanh\left(\frac{q_s}{2}\right)e^{-\frac{iq_s b}{2\pi}(\nu_s + \pi m_-)}$$

$$\times \prod_{s=N+1}^{N+M}\delta\left(2\pi i(M+N-s+\frac{1}{2}) + q_s\right)e^{-\frac{iq_r b m_-}{2}} \times \sum_{\sigma \in S_{N+M}}(-1)^\sigma \prod_{t=1}^{N+M} e^{\frac{iq_t b\mu_{\sigma(t)}}{2\pi}}.$$

$$(A.15)$$

Now it is straightforward to do the $\mu$ and $\nu$ integrals as they have become Gaussian. Doing in

addition some massaging of the expression, we can write it as

$$\mathcal{Z} = \mathcal{N}_4 e^{\frac{MN}{2} Q \pi m_-} e^{\frac{i\pi}{6k} M(2M^2+1)}$$

$$\times \sum_{\sigma \in S_{N+M}} (-1)^\sigma \int d^N p d^{M+N} q \prod_{j=1}^{N} \tanh\left(\frac{p_j}{2}\right) \tanh\left(\frac{q_s}{2}\right) e^{-\frac{im_-}{2}\left(\frac{p_j}{b}+q_j b\right)} e^{\frac{i}{2\pi k} p_j (q_{\sigma(j)} - q_j)}$$

$$\prod_{j=N+1}^{N+M} e^{\frac{1}{k}(M+N-j+\frac{1}{2})q_{\sigma(j)}} \delta\left(2\pi i (M+N-j+\frac{1}{2}) + q_j\right)$$

. (A.16)

Looking at this expression it is clear that the condition $m_- = 0$ is sufficient to guarantee squashing independence. Unfortunately we have not yet managed to show that this is also a necessary condition.

## A.2. The $N_f$ model

Finally let us discuss the simplification of the partition function for the so-called $N_f$ model which consists of an $\mathcal{N} = 4$ vector multiplet, an adjoint hypermultiplet and $N_f$ fundamental hypermultiplets with $U(N)$ gauge group [31][9]. The theory has an $SU(2) \times SU(N_f)$ flavor symmetry which can be deformed by turning on the hypermultiplet mass parameters $m_{\text{adj}}$ and $\vec{m}$ in the Cartan of $SU(2)$ and $SU(N_f)$ respectively. Additionally, we have the $\mathcal{N} = 4$ breaking mass $\mu$ as in eq. (A.8) that we will tune to special values. The full one-loop determinant depends on $(b, m_{\text{adj}}, m_i, \mu)$ where $m_i = \lambda_i(m)$ and $\lambda_i$ is the $i$-th weight of the fundamental representation of $SU(N_f)$. As discussed the one-loop partition function simplifies for different choices of $\mu$

$$Z\left(b, m_{\text{adj}}, m_i, \mu = \pm i \frac{b - b^{-1}}{2}\right) = \left(\frac{b^{\mp 1}}{\cosh\left(\pi b^{\mp 1} m_{\text{adj}}\right)}\right)^N$$

$$\times \prod_{\alpha \in \Delta_+} \frac{\sinh\left(\pi b^{\mp 1} \langle \sigma, \alpha \rangle\right)^2}{\langle \sigma, \alpha \rangle^2 \cosh\left(\pi b^{\mp 1}(\langle \sigma, \alpha \rangle + m_{\text{adj}})\right)^2} \prod_{\alpha \in F} \prod_{i=1}^{N} \frac{1}{\sinh\left(\pi b^{\mp 1}(\langle \sigma, \alpha \rangle + m_i)\right)^2}, \quad (A.17)$$

$$Z\left(b, m_{\text{adj}}, m_i, \mu = +i \frac{b + b^{-1}}{2}\right) = \prod_{\alpha \in \Delta_+} \frac{1}{\langle \sigma, \alpha \rangle}$$

In particular, the first choice of mass-deformation on the squashed sphere gives a partition function which is squashing independent after rescaling $m_{\text{adj}} \to b^{\pm 1} m_{\text{adj}}$, $m_i \to b^{\pm 1} m_i$ and is then equal to the round sphere partition function [32].

---

[9]We thank Shai Chester for suggesting this model to us.

## B. Twisted dimensional reduction of 4d $\mathcal{N} = 2$ conformal supergravity backgrounds

In this appendix we perform a twisted dimensional reduction of four-dimensional supersymmetric backgrounds that arise by taking the rigid limit of the $\mathcal{N} = 2$ conformal supergravity. The resulting three-dimensional backgrounds can then be coupled naturally to three-dimensional $\mathcal{N} = 4$ SYM. We use the three-dimensional backgrounds arising in this twisted dimensional reduction to show that at special values of the chiral multiplet masses on the squashed sphere enhanced supersymmetry is responsible for squashing independence of the free energy.

The field content of the 4d $\mathcal{N} = 2$ Weyl multiplet is [33]

$$g_{MN} \qquad b_M \qquad A_M^r \qquad V_{Mj}^i \qquad \psi_M^i \qquad B_{MN} \qquad \chi^i \qquad D_{4d}. \tag{B.1}$$

The first five fields correspond to the gauge fields for the translations, dilations, $U(1)_r \times SU(2)_R$ R-symmetry and the supersymmetry which resides in the $\mathcal{N} = 2$ superconformal algebra. The scalar $D_{4d}$, the two-form $B_{MN}$ and the dilatino $\chi^i$ are the auxliary fields. The $\mathcal{N} = 2$ SCA also contains Lorentz symmetry, conformal supersymmetry and special conformal transformations whose gauge fields are not independent and are determined in terms of the fields given above. We restrict to supersymmetric backgrounds with vanishing $b_M$, $B_{MN}$ and

$$V_{Mj}^i = \frac{\mathrm{i}}{2} \sigma_{3j}^i A_M^R. \tag{B.2}$$

The variation of gravitino and dilatino is then given by

$$\begin{aligned}
\delta\psi_M^1 &= \nabla_M \xi^1 - \frac{\mathrm{i}}{2} \left( A_M^r + A_M^R \right) \xi^1 - \gamma_M \eta^1, \\
\delta\psi_M^2 &= \nabla_M \xi^2 - \frac{\mathrm{i}}{2} \left( A_M^r - A_M^R \right) \xi^2 - \gamma_M \eta^2, \\
\delta\chi^1 &= D_{4d} \xi^1 + \frac{\mathrm{i}}{3} \gamma^{MN} \xi^1 \partial_M \left( -\frac{1}{2} A_N^R + A_N^r \right), \\
\delta\chi^2 &= D_{4d} \xi^2 + \frac{\mathrm{i}}{3} \gamma^{MN} \xi^2 \partial_M \left( \frac{1}{2} A_N^R + A_N^r \right),
\end{aligned} \tag{B.3}$$

where $\eta^{1,2}$ are arbitrary spinors. Supersymmetric theories can be placed on a manifold if the above variations vanish for a choice of background fields and some non-trivial $\xi^{1,2}$. It is convenient to organize the gauge fields $A^r, A^R$ appearing above in terms of the gauge field for the symmetry $R_0$ which only rotates the fermions and the flavor symmetry of an $\mathcal{N} = 1$ subalgebra

$$r A_M^{\mathrm{r}} + R A_M^{\mathrm{R}} = \frac{R_0}{2} \left( A_\mu^{\mathrm{R}} + A_\mu^{\mathrm{r}} \right) + F A_\mu^{\mathrm{r}} \equiv F A_M^{\mathrm{F}} + R_0 A_M^{\mathrm{R}_0} \tag{B.4}$$

The spinor $\xi^1$ is not charged under the flavor symmetry while the spinor $\xi^2$ is charged under both.

The conditions for a supersymmetric background can then be written as

$$\nabla_M \xi^1 - \mathrm{i} A_M^{\mathrm{R_0}} \xi^1 = \gamma_M \eta^1,$$
$$\nabla_M \xi^2 + \mathrm{i} \left( A_M^{\mathrm{R_0}} - A_M^{\mathrm{F}} \right) \xi^2 = \gamma_M \eta^2,$$
$$D_{\mathrm{4d}} \xi^1 - \frac{\mathrm{i}}{3} \gamma^{MN} \xi^1 \partial_M \left( A_N^{\mathrm{R_0}} - \frac{3}{2} A_N^{\mathrm{F}} \right) = 0,$$
$$D_{\mathrm{4d}} \xi^2 + \frac{\mathrm{i}}{3} \gamma^{MN} \xi^2 \partial_M \left( A_N^{\mathrm{R_0}} + \frac{1}{2} A_N^{\mathrm{F}} \right) = 0.$$

$$(\mathrm{B}.5)$$

By dimensional reduction, we now obtain the equations for a supersymmetric background in three dimensions. We start with the four dimensional metric

$$\mathrm{d}s_{\mathrm{4d}}^2 = G_{MN} \mathrm{d}x^M \mathrm{d}x^N = \left( \mathrm{d}x^4 + c_\mu \mathrm{d}x^\mu \right)^2 + g_{\mu\nu} \mathrm{d}x^\mu \mathrm{d}x^\nu \tag{B.6}$$

and frames (small latin indices denote the $3d$ frame indices)

$$E^4 = \mathrm{d}x^4 + c_\mu \mathrm{d}x^\mu, \qquad E^a = e_\mu^a \mathrm{d}x^\mu \tag{B.7}$$

and the inverse

$$E_4 = \frac{\partial}{\partial x^4}, \qquad E_a = e_a{}^\mu \frac{\partial}{\partial x^\mu} - e_a{}^\mu c_\mu \frac{\partial}{\partial x^4}. \tag{B.8}$$

The spin connections are related by

$$\Omega_{\mu ab} = \omega_{\mu ab} + \frac{1}{2} c_\mu e_a^\nu e_b^\rho \left( \partial_\nu c_\rho - \partial_\rho c_\nu \right),$$
$$\Omega_{\mu 4b} = \frac{1}{2} e_b^\nu \left( \partial_\mu c_\nu - \partial_\nu c_\mu \right),$$
$$\Omega_{4ab} = \frac{1}{2} e_b^\nu e_b^\rho \left( \partial_\nu c_\rho - \partial_\rho c_\nu \right),$$
$$\Omega_{44b} = 0.$$

$$(\mathrm{B}.9)$$

The covariant derivatives of a spinor $\chi_\alpha$ which does not depend on $x^4$ are related by

$$\nabla_\mu^{\mathrm{4d}} \chi = \nabla_\mu + \frac{1}{2} \varepsilon_{\mu\nu\rho} V^\nu \gamma^\rho \chi + \frac{1}{2} c_\mu V^\nu \gamma_\nu \chi, \qquad \nabla_4 \chi = \frac{1}{2} V^\mu \gamma_\mu \chi, \tag{B.10}$$

where $V^\mu = -\frac{\mathrm{i}}{2} \varepsilon^{\mu\nu\rho} \partial_\nu c_\rho$ and our conventions for the gamma matrices is

$$\gamma_{\alpha\dot\alpha}^M = (\vec\sigma, -\mathrm{i}), \qquad \widetilde\gamma^{M\dot\alpha\alpha} = (-\vec\sigma, -\mathrm{i}) \tag{B.11}$$

Let us start with a generic equation of the form

$$\left( \nabla_M - \mathrm{i} A_M \right) \xi = \gamma_M \widetilde\eta. \tag{B.12}$$

Using the $M = 4$ component of the equation we obtain

$$\widetilde{\eta} = \frac{i}{2} V^\mu \gamma_\mu \xi + A_4 \xi. \tag{B.13}$$

Using this in the $M = \mu$ component along with the gamma matrices we get

$$\left( \nabla_\mu - i \left( A_\mu + \frac{1}{2} V_\mu \right) - A_4^{4d} \gamma_\mu + \varepsilon_{\mu\nu\rho} V^\nu \gamma^\rho \right) \xi = 0, \tag{B.14}$$

where $A_\mu = A_\mu^{4d} - c_\mu A_4^{4d}$ is the three dimensional one-form and $A_4^{4d}$ is a three dimensional scalar. Applying this result to $\delta\psi_M^{1,2} = 0$ we obtain

$$\left( \nabla_\mu - i \left( A_\mu^{R_0} + \frac{1}{2} V_\mu \right) - H \gamma_\mu + \varepsilon_{\mu\nu\rho} V^\nu \gamma^\rho \right) \xi^1 = 0$$
$$\left( \nabla_\mu + i \left( A_\mu^{R_0} - A_\mu^{F} - \frac{1}{2} V_\mu \right) - (\sigma - H) \gamma_\mu + \varepsilon_{\mu\nu\rho} V^\nu \gamma^\rho \right) \xi^2 = 0, \tag{B.15}$$

where $H = A_4^{R_0}$ and $\sigma = A_4^{F}$. The dilatino variations are of the form

$$D_{4d}\epsilon + i\gamma^{MN}\epsilon\partial_M A_N^{4d}, \tag{B.16}$$

which upon dimensional reduction becomes

$$(D + \sigma H)\epsilon + \varepsilon^{\mu\nu\rho}\partial_\mu A_\nu \gamma_\rho \epsilon + \gamma^\mu \epsilon (\partial_\mu + 2iV_\mu) A_4. \tag{B.17}$$

Using this the two dilatino variations become

$$\delta\chi^1 = (D + \sigma H)\xi^1 - \frac{1}{3}\left( \varepsilon^{\mu\nu\rho}\gamma^\rho \xi^1 \partial_\mu \left( A_\nu^{R_0} - \frac{3}{2} A_\nu^{F} \right) + \gamma^\mu \xi^1 (\partial_\mu + 2iV_\mu)\left( H - \frac{3}{2}\sigma \right) \right) = 0,$$
$$\delta\chi^2 = (D + \sigma H)\xi^2 + \frac{1}{3}\left( \varepsilon^{\mu\nu\rho}\gamma^\rho \xi^2 \partial_\mu \left( A_\nu^{R_0} + \frac{1}{2} A_\nu^{F} \right) + \gamma^\mu \xi^2 (\partial_\mu + 2iV_\mu)\left( H + \frac{1}{2}\sigma \right) \right) = 0. \tag{B.18}$$

## C. Extra spinors in ABJ(M)

In the main text we have shown that special values of masses correspond to supersymmetry enhancement by embedding the $3d$ $\mathcal{N} = 2$ supergravity equations in a twisted dimensional reduction of $4d$ $\mathcal{N} = 2$ supergravity. This explains the simplfication of free energy for $3d$ theories which can be obtained by a dimensional reduction of $4d$ theory. ABJ(M) theories, however, cannot be obtained in such a way so the previous arguments do not apply. In this section we consider the $3d$ $\mathcal{N} = 6$ conformal supergravity with matter coupling, whose rigid limit gives ABJ(M) theories on supersymmetric background. We show that our conclusions regarding the symmetry enhancement at special values of mass hold.

The field content of the $3d$ $\mathcal{N}=6$ Weyl multiplet is

$$g_{\mu\nu} \qquad A_\mu^{IJ} \qquad A_\mu \qquad E^{IJ} \qquad D^{IJ} \qquad \psi_\mu^I \qquad \chi^{IJK}, \qquad \chi^I \tag{C.1}$$

where $I, J, K, \cdots = 1, 2, \cdots, 6$. $A_\mu^{IJ}$ and $A_\mu$ are gauge fields for the $SO(6) \times U(1)$ R-symmetry. $E^{IJ}$ and $D^{IJ}$ are scalar fields in the **15** of $SO(6)$. $\chi^{IJK}$ and $\chi^I$ are dilatinos. The general conditions for supersymmetric background are obtained by setting SUSY variations of all fermions to zero. These variations are given in detail in eq. (2.5) of [25].

When $\mathcal{N}=6$ ABJ(M) is coupled to this conformal supergravity then the fields $E_{IJ}$ and $D_{IJ}$ appear in the general mass matrix for scalars and fermions. In the $\mathcal{N}=2$ language, turning on supersymmetric background fields for the Cartans of $SO(6)$ flavor symmetry correspond to turning on constant $12, 34$ and $56$ components of $E^{IJ}$ and $D^{IJ}$. From the SUSY variations of fermions it is clear that in the massless case two spinors are preserved, those corresponding to say $I = 1, 2$. This is because of the fact that one need to turn on background R-fields to preserve supersymmetry. If only background R-fields are turned on then supersymmetric variations of various fermions in the supergravity multiplet can't be set to zero unless some of the spinors vanish identically. This manifestly preserves only one-third of the supersymmetry (i.e., four supercharges).

Next we show that the supersymmetry can be enhanced to eight and six supercharges by turning on a special mass. Without loss of generality we start with spinors $\epsilon^1$ and $\epsilon^2$ preserved which we combine in complex combinations $\xi^1$ and $\widetilde{\xi}^1$ which satisfy the conformal Killing spinor equation

$$\nabla_\mu \xi^1 - \mathrm{i} A_\mu^{12} \xi^1 = \gamma_\mu \eta^1, \qquad \eta^1 = H\xi^1 + \mathrm{i} V^\mu \gamma_\mu \xi^1, \qquad A_\mu^{12} = -V_\mu, \tag{C.2}$$

for $V, H$ as in eq. (3.4) and $\xi^1$ as in eq. (2.20). In addition, we want to preserve the spinors $\epsilon^{3,4}$ by turning on scalars $E_{56}$ and $D_{56}$ and some gauge fields for the R-symmetry in the supergravity multiplet. We again combine these spinors into complex combinations. The relevant variations of fermions that are not trivially zero are given by

$$\delta\left(\psi_\mu^3 + \mathrm{i}\psi_\mu^4\right) = \nabla_\mu \xi^2 - \mathrm{i} A_\mu^{34} \xi^2 - \gamma_\mu \eta^2 = 0$$

$$\delta\left(\lambda^{123} + \mathrm{i}\lambda^{124}\right) = \frac{1}{4\sqrt{2}}\gamma^{\mu\nu}G_{\mu\nu}^{12}\xi^2 - \mathrm{i}\left(D^{56}\xi^2 - E^{56}\eta^2\right), \tag{C.3}$$

$$\delta\left(\lambda^{341} + \mathrm{i}\lambda^{342}\right) = \frac{1}{4\sqrt{2}}\gamma^{\mu\nu}G_{\mu\nu}^{34}\xi^1 - \mathrm{i}\left(D^{56}\xi^1 - E^{56}\eta^1\right).$$

To preserve the $\mathcal{N}=2$ supersymmetry, we see that a non-trivial $A_\mu^{34}$ needs to be turned on:

$$D^{56} = HE^{56} = \frac{\mathrm{i}}{4r_3}\left(b + b^{-1}\right)E^{56}, \qquad A_\mu^{34} = 2\mathrm{i}\sqrt{2}E^{56}\frac{r_3}{b + b^{-1}}V_\mu. \tag{C.4}$$

The spinors $\xi^2$ are also preserved if we choose the flavor parameter such that $A_\mu^{34} = -V_\mu$,

this corresponds to when $E^{56}$ takes the value $E^{56*} = \frac{\mathrm{i}(b+b^{-1})}{r_3}\frac{1}{2\sqrt{2}}$. Up to an overall constant this corresponds to the second squashing independent point for ABJ(M) discussed below. In this case four extra supercharges are preserved and the Killing $\xi^2$ is actually equal to $\xi^1$.

Similarly, for the other squashing independent point, we take $\xi^2$ to be an eigenspinor of $\gamma^3$. We then use the second equation in eq. (C.3) to determine $\eta^2$

$$\eta^2 = H\xi^2 + \mathrm{i}\frac{E^{56*}}{E^{56}}V^\mu\gamma_\mu\xi^2. \tag{C.5}$$

Using this the Killing spinor equation for $\xi^2$ becomes

$$\nabla_\mu\xi^2 + \mathrm{i}\frac{E^{56}}{E^{56*}}V_\mu\xi^2 - H\gamma_\mu\xi^2 - \mathrm{i}\frac{E^{56*}}{E^{56}}V^\nu\gamma_\mu\gamma_\nu\xi^2 = 0. \tag{C.6}$$

For $\xi^2 = (1,0)$ we have $V^\mu\gamma_\mu\xi^2 = -\frac{b-b^{-1}}{2r_3}\xi^2$. If $E^{56} = \mathrm{i}\left(\frac{b-b^{-1}}{2\sqrt{2}r_3}\right)$ then the above equation becomes

$$\nabla_\mu\xi^2 + \mathrm{i}\frac{b-b^{-1}}{b+b^{-1}}V_\mu\xi^2 + \frac{\mathrm{i}}{4}\frac{b+b^{-1}}{r_3}\gamma_\mu\xi^2 = 0, \tag{C.7}$$

which is indeed satisfied for the constant spinor. Note that these are precisely the squashed sphere Killing spinors discussed in section 2.1 for which the partition function is known to be squashing independent.

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
