# Peer review of "Squashing and supersymmetry enhancement in three dimensions"

_SciPost Physics_

## Round 1 · Referee Report · Anonymous (Referee 1) · 2021-8-19

Strengths

  1. Gives a clear geometrical explanation for a recently observed simplification of the squashed sphere free energy for ABJM theory.

Report

This paper explains a recently observed demonstration of the simplification of the mass deformed squashed sphere free energy in ABJM theory when the mass is fixed to a certain function of squashing. The explanation is that supersymmetry is enhanced specifically at this point, which the authors show explicitly. They also show a similar simplification at a different value of the mass, and also similar simplifications for 3d CFTs with less supersymmetry.

Requested changes

  1. on page 1, it says "Using holographic duality, these relations imply non-trivial constraints on string theory amplitudes and can be used to rule out certain higher-derivative terms in string theory effective actions". I would more accurately say that the constraints allow one to fix protected terms in the effective action, which in some case turn out to be zero (but in other cases are nonzero).

  2. on page 2, it says "A similar phenomenon was discovered by [17] for ABJM theory". This relation was actually shown for the more general case of ABJ theory (ie nonequal ranks of the gauge group). In general throughout the paper, it would be useful to generalize results to the case of ABJ, which should be straightforward.

  3. it would be useful to reference the results of appendix A.2 for the Nf model in the main text, as otherwise readers might not be aware of it.

---

## Round 1 · Referee Report · Anonymous (Referee 2) · 2021-11-8

Report

In this paper, the authors consider mass-deformed 3d ${\cal N} = 2$ theories on a squashed sphere, and they determine certain relations between the mass and the squashing parameter at which there is more supersymmetry. In the cases in which there is supersymmetry enhancement, the authors point out that the $S^3$ partition function simplifies, and in some cases it even becomes independent of the squashing parameter, as had been observed in previous work.

I think the paper is very interesting, and I have a few small comments about potential improvements:

  1. When examining the ellipsoid in section 2.2, it would be very useful to write down the frame in which the spinors are given.

  2. Similarly, it would be useful to write down the gamma matrices that are used throughout the paper, and also explain the conventions used for multiplying spinors (for instance, in (2.6)).

  3. I suspect that in equation (3.3) the right-hand sides of the two lines were accidentally swapped. The authors should double check whether the equations are correct as written.

  4. Regarding (A.6): for 3d theories with ${\cal N} = 4$ supersymmetry, one can have vector multiplets and hypermultiplets, but one can also have twisted vector multiplets and twisted hypermultiplets. It would be useful to give the partition functions of the latter two as well, in case they are different from the ones for vector multiplets and hypermultiplets, respectively.

  5. Actually, the authors should check whether the formulas in (A.6) are correct. I would've thought that the second line should either have only $\cosh (\pi b \langle \sigma, \alpha \rangle )$ or $\cosh (\pi b^{-1} \langle \sigma, \alpha \rangle )$, with a product over all roots, but I may be wrong. (In other words, I would've thought that this formula should not be symmetric under $b \to 1/b$.)

  6. It wasn't clear to me whether the "explaination" of the fact that the partition function of ABJM theory is independent of squasing (provided that the mass is tuned such that there is enhanced supersymmetry) is complete. The authors have shown that there is enhanced supersymmetry, but what I think is still needed is to show that the squashing deformation is Q-exact in this case. It would be great if the authors could add some clarifying comments about whether this Q-exactness is obvious from their analysis, or whether there is still some work to be done.

In any case, I think this paper is of interest to the community of researchers studying supersymmetric field theories on curved manifolds, so I recommend it for publication after the authors have had a chance to consider the improvements/comments mentioned above.

---

## Round 2 · Author Response

We thank the referees for their careful reading of the manuscript and for their valuable suggestions. We have made several changes in the new version following the points made by the referees. These changes are listed in the the appropriate section of our resubmission page.

---

## Round 2 · List of Changes

1. We use more precise language on page 1 as advocated by the first referee.
2. We generalized our discussions so that they are now applicable to ABJ(M) theory.
3. We mention the $N_f$ model at the end of paragraph 5 of the introduction, as well as in the last paragraph of the introduction.
4. We added the frame for the ellipsoid as eq. (2.29).
5. We added a footnote on the gamma matrix convention to equation (2.1).
6. We added a comment on twisted multiplets in the paragraph before (A.6).
7. We fixed typos and changed the notation in appendix A to clarify various expressions. The simplified one-loop determinants for the $N=4$ vector and hyper multiplets are now given in eq. (A.6) through (A.9).
8. In the second to last paragraph of the introduction we have outlined the logic of the paper which addresses the referee’s remark regarding Q-exactness.

Apart from the above changes directly related to the referees’ concerns we have taken the opportunity to fix several typos and make improvements to the draft. The significant changes are listed below.

1. We added a footnote on contracting spinors to eq. (2.6).
2. We added a note on $\mu=-\io\frac{b+b^{-1}}/2$ after (4.20).
3. In (A.3) and the following equations we corrected the charge that couples to $\mu$ as the  flavor charge $F$.
4. We added expressions for the ${\cal N}=4$ vector one-loop partition function, explaining the squashing independence.
5. We added a comment on the flavor charge of the ${\cal N}=4$ multiplet constituents.
6. We revised the $N_f$-model in section (A.2).

You are currently on this page

Resubmission 2107.07151v2 on 17 November 2021

---

## Editorial Decision

publication_decision_taken:_accept